# Temporally Consistent Atmospheric Turbulence Mitigation with Neural Representations

**Haoming Cai**[*1], **Jingxi Chen**[*1], **Brandon Y. Feng**[2], **Weiyun Jiang**[3], **Mingyang Xie**[1],
**Kevin Zhang**[1], **Cornelia Fermuller**[1], **Yiannis Aloimonos**[1], **Ashok Veeraraghavan**[3],
**Christopher A. Metzler**[†1]

[1]University of Maryland, [2]Massachusetts Institute of Technology, [3]Rice University

## Abstract

Atmospheric turbulence, caused by random fluctuations in the atmosphere's refractive index, introduces complex spatio-temporal distortions in imagery captured at long range. Video Atmospheric Turbulence Mitigation (ATM) aims to restore videos affected by these distortions. However, existing video ATM methods, both supervised and self-supervised, struggle to maintain temporally consistent mitigation across frames, leading to visually incoherent results. This limitation arises from the stochastic nature of atmospheric turbulence, which varies across space and time. Inspired by the observation that atmospheric turbulence induces high-frequency temporal variations, we propose ConVRT, a novel framework for consistent video restoration through turbulence. ConVRT introduces a neural video representation that explicitly decouples spatial and temporal information into a spatial content field and a temporal deformation field, enabling targeted regularization of the network's temporal representation capability. By leveraging the low-pass filtering properties of the regularized temporal representations, ConVRT effectively mitigates turbulence-induced temporal frequency variations and promotes temporal consistency. Furthermore, our training framework seamlessly integrates supervised pre-training on synthetic turbulence data with self-supervised learning on real-world videos, significantly improving the temporally consistent mitigation of ATM methods on diverse real-world data. More information can be found on our project page: https://convrt-2024.github.io/

## 1 Introduction

Atmospheric turbulence poses a significant challenge in long-range imaging applications, causing unique distortions in captured videos. These turbulence-distorted videos suffer from spatially-varying and time-varying degradations, including blur and warping effects, due to the random fluctuations of the refractive index in the atmosphere. These distortions significantly hinder the performance of computer vision applications like object detection, recognition, and surveillance systems by obscuring the true shapes, edges, and visual details of objects. Therefore, this work focuses on Video Atmospheric Turbulence Mitigation (ATM), aiming to recover videos degraded by these atmospheric distortions.

Mathematically, the process of capturing video through atmospheric turbulence can be modeled by the following equation

---

*Equal contributions † Corresponding author.

38th Conference on Neural Information Processing Systems (NeurIPS 2024).

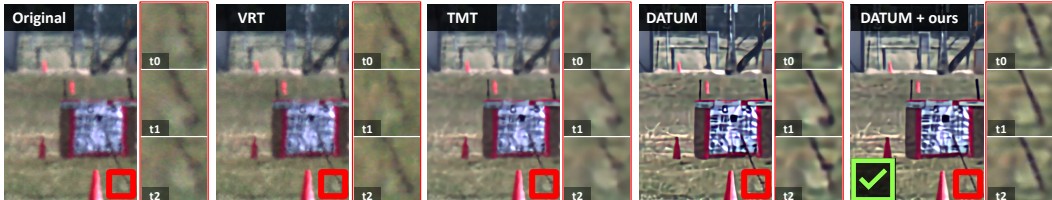

Figure 1: **Temporally consistent restoration in video ATM is challenging.** State-of-the-art methods like DATUM (2) (CVPR'24) and TMT (3)(TCI'23), designed for video ATM, fail to maintain temporal consistency in real-world atmospheric turbulence. For instance, they produce flickering artifacts on a static pole.

$$\underset{\text{distorted frame}}{y_t} = \underbrace{[B_t \circ T_t]}_{\text{tilt-then-blur } \mathcal{H}} \left( \underset{\text{clean frame}}{I_t} \right), \tag{1}$$

where $B_t$ and $T_t$ are blur and tilt process at t time stamp. $\circ$ denotes the application of tilt followed by blur (1).

As described by Equation (1), the key challenge arises from the stochastic nature of atmospheric turbulence, which varies across space and time, making temporally consistent video restoration difficult. Figure 1 illustrates the challenge of temporal inconsistency by showing a static scene captured with a stationary camera and object. Despite the static setup, the atmospheric turbulence introduces erratic movements of the stationary traffic cone across frames, causing flickering artifacts in the resulting video sequence. Notably, even state-of-the-art methods like DATUM (2), designed for video turbulence mitigation, fail to produce temporally consistent mitigation results, result in flickering video. This underscores the critical need for novel solutions tailored to address the challenge of temporal consistency in video atmospheric turbulence mitigation.

## 1.1 Current State-of-the-Art in Atmospheric Turbulence Mitigation

To address this complex and variable degradation in images and videos, various methodologies have been developed. Current state-of-the-art methods can generally be categorized into supervised and self-supervised learning manner.

Supervised learning techniques in ATM use turbulence simulators to generate paired training data (clean and distorted images/video) that can be used for training (14; 15; 16; 1; 17; 18). Fig.3(A) and the supervised learning section of Table 1 depict methods that achieve significant results based

Table 1: Comparison of recent supervised (S), self-supervised (SS), and hybrid (S+SS) learning approaches for image and video ATM.

| Supervision | Method | Capability | Critical Performance Factors |
|---|---|---|---|
| | TSRWGAN (4) | Static Scene Sequences | Adversarial Learning |
| | TurbNet (5) | Image | Advanced Simulator |
| S | PiRN-SR (6) | Image | Advanced Simulator |
| | TMT (7) | Video | Physically-Grounded Model |
| | DATUM (2) | Video | Physically-Grounded Model |
| | Turb-Seg-Res (8) | Video | Advanced Simulator |
| | Mao et al. (9) | Image | Lucky Imaging & Denoisers |
| | Li et al. (10) | Image | Degradation Est |
| SS | TurbuGAN (11) | Static Scene Sequences | Adversarial Sensing Concept |
| | NeRT (12) | Static Scene Sequences | Degradation Est |
| | Diff. Template (13) | Static Scene Sequences | Optical Flow |
| S+SS | ConVRT (ours) | Video | Representation Regularization |

on large amounts of paired data. Despite the continual evolution of simulators, the persistent gap between simulated and real-world atmospheric turbulence poses challenges for this design in handling unseen real-world data. In videos, this drawback is further amplified, leading to issues like temporally inconsistent mitigation. To address this temporal inconsistency issue, in addition to better simulators, enlarging the dataset and model capacity are necessary, which substantially increases the computational costs of training.

Self-supervised learning approaches for ATM employ internal learning techniques to leverage data priors such as lucky images, internal data distributions, or blind degradation estimation, as depicted in Fig. 3(B) and self-supervised learning section of Table 1. A key advantage of these methods is their test-time optimization capability, allowing them to adapt to any test data. However, to date these approaches have not been used to enforce temporal consistency in video ATM. Furthermore,

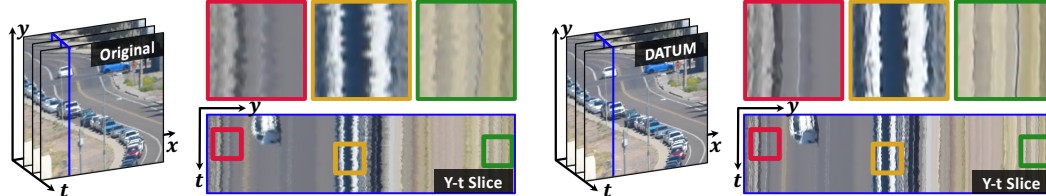

Figure 2: **Inspiration of our method:** Atmospheric turbulence introduces high-frequency temporal variations in videos due to the chaotic motion of air caused by temperature gradients and other energy sources. These variations manifest as time-varying tilt and blur, deviating from the ground truth, as evident in the rapidly fluctuating patterns along the temporal dimension (vertical axis) of the y-t slice of the turbulence-distorted video (left). In contrast, the restored video of SOTA method DATUM (2) (right) exhibits smoother temporal variations, indicating the mitigation of turbulence-induced distortions. This key insight highlights the potential for regularizing temporal information to effectively restore videos affected by atmospheric turbulence.

because they don't exploit accurate learned image priors, the performance of self-supervised methods in real-world turbulence mitigation often falls short of supervised learning approaches.

This paper develops a hybrid algorithm that can consistently mitigate real-world atmospheric turbulence across video frames. Our pipeline can leverage the knowledge encoded in pre-trained models while leveraging test-time optimization to adapt to the complexities of real-world turbulence. As shown in Fig. 3(C) and the final section of Table 1, our pipeline leverages the strengths of both self-supervised learning and simulation-based pre-training.

### 1.2 Motivation and Contribution

Our work is motivated by the insight, illustrated in Figure 2, that state-of-the-art ATM methods struggle to remove the temporal distortions introduced by turbulence. That is, while existing methods are reasonably effective at removing the spatial distortions (e.g., blur) introduced by turbulence, they do not effectively remove the temporal distortions.

To address this challenge, we develop an approach that explicitly decouples spatial and temporal information. This method leverages the low-pass filtering properties of neural networks to reduce turbulence-induced degradations. Specifically, we propose a self-supervised method called ConVRT (**Con**sistent **V**ideo **R**estoration through **T**urbulence). ConVRT forms a neural representation of the reconstucted video that explicitly decouples spatial and temporal information: The video is represented with a spatial content field and a temporal deformation field. This decoupling allows ConVRT to effectively regularize the temporal information while preserving spatial information and fine details.

Through extensive evaluations, we demonstrate that ConVRT substantially improves temporally consistency while also marginally improving per-frame restoration quality.

## 2 Related Work

**Implicit neural representations.** Our work leverages a coordinate-based implicit neural representation (INRs), which has been commonly adopted to model 2D images or 3D videos as multi-layer perceptions (MLPs). INRs take 2D pixel coordinates $(x, y)$, or 3D pixel coordinates with temporal encoding, $(x, y, t)$ and output the corresponding pixel values. These INRs demonstrate exceptional performance when fitting images (19; 20; 21; 22; 23; 24; 25), videos (26; 22), 3D shapes (27; 28; 26; 29; 30; 31), and optical components (32). Not only they are able to represent these 2D or 3D signals, but they also show strong priors for solving inverse problems, such as image super resolution (33), phase retrieval (34), and reducing optical aberration (35; 36; 37; 38).

**Neural video representation.** Our work aligns closely with the evolving field of neural video representation (39; 40; 41; 42). While there are existing approaches (43; 44; 42; 45) that seek to represent a video into decomposed layers, these primarily focus on clean videos and are not applicable to videos with severe degradation turbulence. Our work extends the application of neural video

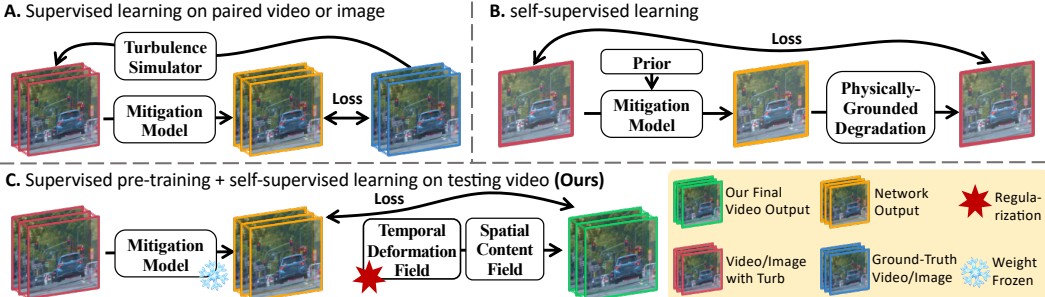

Figure 3: Comparison of different learning approaches for image and video ATM. (a) ATM methods in supervised learning face challenges in domain adaptation for real-world data. (b) Self-supervised learning ATM methods mostly explore static video sequences, outputting one frame from multiple frames as input. (c) Our hybrid pipeline is tailored for video ATM, combining supervised pre-training and self-supervised learning to achieve consistent video restoration through turbulence

representation to scenarios heavily impacted by atmospheric turbulence. This extension is not trivial, as it involves addressing the unique challenges posed by the dynamic and unpredictable nature of turbulence, which are not considered in conventional video representations.

**Atmospheric turbulence mitigation.** Attempts to mitigate atmospheric turbulence (46; 47) have applied optical flow (9; 48), B-spline grid (49), and diffeomorphism (50) to unwarp each distorted image and then fuse and combine these registered distorted images into a clean and sharp image. The fusion is usually modeled as patch-wise stitching (9) or blind deconvolution (51). Recent development of high-performance GPUs and fast turbulence simulators (16; 18; 17; 15; 14) leads to new progress in turbulence mitigation (15; 5; 11; 7; 12). However, previous efforts tend to overlook the importance of temporal consistency on the reconstructed video. Our method, ConVRT, is specifically designed to restore temporal consistency with on test-time optimization of a neural video representation.

**Blind Video Restoration via deep video prior.** Supervised video restoration methods (52; 53; 54; 55) have made significant advancements but are constrained by the need for paired data, which increases the value of blind video restoration. One promising direction involves leveraging deep priors. The deep video prior (DVP) and DVP-based blind video consistency methods (56; 57) use convolutional neural networks (CNNs) to learn image operators that exploit the implicit priors in CNNs to remove video artifacts. These approaches have demonstrated impressive results in tasks such as colorization and white-balancing. However, turbulence mitigation presents a more complex challenge compared to these common degradations, involving spatially and temporally varying blur and tilt. This complexity raises unexplored questions for these types of methods

## 3 Method

### 3.1 Overview of the Pipeline

The framework of our method, ConVRT, is presented in Figure 4. In this subsection, we provide a high-level overview covering the design inspiration, the video representation mechanism, and the training process.

**Design Inspiration.** As discussed in Section 1.2, the core design logic of ConVRT is to apply temporal-wise regularization in video representation learning. For the representation, our method is inspired by a series of works on tensor decomposition, commonly used to parameterize 3D volumes in implicit neural representations (INR). These approaches enhance the ability to represent 3D signals while reducing the number of required parameters (33; 58; 59). Building upon this, we developed the ConVRT method.

**Video Representation.** ConVRT represents videos using two main components: the 3D Spatial-Temporal Deformation Field ($T_{\text{field}}$) and the 2D Spatial Content Field ($S_{\text{field}}$). The process begins with $T_{\text{field}}$, which receives the pixel location $(x, y, t)$ as input, where $(x, y)$ are spatial coordinates and $t$ is the temporal frame index. $T_{\text{field}}$ outputs deformation offsets $(\Delta x, \Delta y)$, indicating changes in the pixel's spatial position across frames relative to a canonical frame. These offsets are then used by

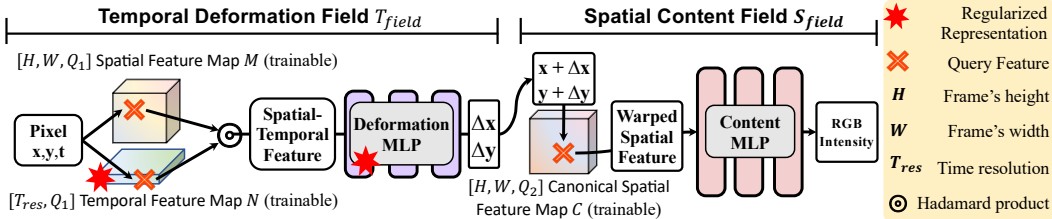

Figure 4: Illustration of the proposed method. ConVRT represents a video with two fields: the Temporal Deformation Field ($T_{field}$) and the Spatial Content Field ($S_{field}$). Regularization is applied by constraining the dimensions of the Temporal Feature Map. Similarly, reducing the size of the deformation MLP serves as additional regularization to promote temporal consistency.

$S_{\text{field}}$. $S_{\text{field}}$ queries the Canonical Spatial Feature Map ($C$) at the modified location $(x + \Delta x, y + \Delta y)$ to retrieve the corresponding feature from a trainable feature map. This feature is subsequently processed by an MLP to predict the RGB intensity values for the pixel location $(x, y, t)$.

**Training Overview.** During training, the trainable parameters include all feature maps and the Multi-Layer Perceptrons (MLP) described below. The loss function measures the difference between the predicted RGB intensity values and the corresponding pixel colors in the restored video, obtained using any ATM method. Since no ground-truth data is available, ConVRT is designed to overfit each partially restored video; however, its limited capacity for capturing temporal information prevents it from overfitting to turbulence artifacts

## 3.2 Temporal Deformation Field $T_{field}$ with Regularization

We represent the input video's spatial-temporal features using two main components: the Spatial Feature Map and the Temporal Feature Map. The Spatial Feature Map ($M$) acts as a dictionary for spatial features, with dimensions $\mathbb{R}^{H \times W \times Q_1}$, where $H$ is the frame height, $W$ is the frame width, and $Q_1$ is the number of spatial feature channels. Each pixel coordinate $(x, y)$ serves as a key to retrieve the corresponding spatial feature vector $M_{x,y} \in \mathbb{R}^{Q_1}$. The Temporal Feature Map ($N$) functions as a dictionary for temporal features, with dimensions $\mathbb{R}^{T_{res} \times Q_1}$. Here, $T_{res}$ is the regularized temporal resolution and $Q_1$ is the number of temporal feature channels.

To construct the spatial-temporal feature vector $V_{x,y,t}$ at a specific pixel location $(x, y, t)$, we first query the Spatial Feature Map $M$ using the pixel coordinates $(x, y)$, extracting the spatial feature vector $M_{x,y} \in \mathbb{R}^{Q_1}$. Next, we query the Temporal Feature Map $N$ using the time coordinate $t$, extracting the temporal feature vector $N_t \in \mathbb{R}^{Q_1}$. These vectors are then combined using the Hadamard product, which performs element-wise multiplication, to form the spatial-temporal feature vector:

$$V_{x,y,t} = M_{x,y} \odot N_t \tag{2}$$

This Hadamard product effectively combines the spatial and temporal features, creating a compact and efficient representation of the video's spatial-temporal characteristics. This $V_{x,y,t}$ is then fed into a compact MLP, referred to as the deformation MLP. The details of the deformation MLP are provided in the supplementary material. The deformation MLP outputs the offsets $(\Delta x, \Delta y)$ necessary for warping the canonical spatial feature map.

To regularize the temporal representation capability of the Temporal Feature Map ($N$), we constrain its dimensions to $\mathbb{R}^{T_{res} \times Q_1}$, where $T_{res}$ is much smaller than the total number of video frames ($T$). Consequently, multiple neighboring frames share the same temporal feature. For example, frames at $t - 1$, $t$, and $t + 1$ may query the same temporal feature $N_t$ due to the reduced temporal resolution. Additionally, we define the deformation MLP with a reduced number of parameters. Both regularizations decrease the representation capacity of the temporal features, promoting smoother and more consistent temporal dynamics across frames, as inspired by our motivation experiment.

### 3.3 Spatial Content Field

The Spatial Content Field focuses on accurately representing the spatial details of each video frame. Unlike the Spatial Feature Map ($M$) used in the Temporal Deformation Field, we initialize a new optimizable feature map, denoted as the Canonical Spatial Feature Map ($C$), with dimensions $\mathbb{R}^{H \times W \times Q_2}$, where $H$ is the frame height, $W$ is the frame width, and $Q_2$ is the number of spatial feature channels specific to this field.

Each pixel coordinate $(x, y)$ is adjusted by the deformation offsets $(\Delta x, \Delta y)$, resulting in new coordinates $(x + \Delta x, y + \Delta y)$. These adjusted coordinates are then used to query the Canonical Spatial Feature Map ($C$), retrieving the spatial feature vector $C_{x+\Delta x, y+\Delta y} \in \mathbb{R}^{Q_2}$. These spatial features are processed by a Content MLP, which transforms the spatial feature vector $C_{x+\Delta x, y+\Delta y}$ into the final RGB intensity values for the corresponding pixel. The details of the Content MLP are provided in the supplementary material. This transformation ensures that the spatial details of the video frame are accurately captured and represented.

### 3.4 Training Objectives

**Temporal Consistency Regularization.** To ensure temporal stability across video frames, we use a disparity estimation network (MiDas (60)) to calculate pixel-wise disparities. These disparities serve as weights for the predicted warp (one of $D_{\text{field}}$'s outputs), helping to maintain spatial consistency over time. The loss is defined as:

$$\mathcal{L}_{temp} = (1 - \text{Disparity}(I)) \cdot \|\text{Predicted Warp}\|_1 \tag{3}$$

where $\text{Disparity}(I)$ measures the pixel-level disparity, and $\|\text{Predicted Warp}\|_1$ enforces sparsity in the grid changes. The design of $\mathcal{L}_{temp}$ minimizes the L1 norm of the predicted warp, conditioned by $1 - \text{Disparity}(I)$, to prioritize consistency in far regions based on the depth information. This focused approach on temporal consistency significantly reduces the propagation of turbulence-induced distortions, ensuring a smooth transition between frames.

**Similarity Loss.** The Similarity Loss Term is given by:

$$\mathcal{L}_{sim} = \lambda_{mse}\mathcal{L}_{mse} + \lambda_{ssim}\mathcal{L}_{ssim} + \lambda_{lpips}\mathcal{L}_{lpips} \tag{4}$$

where $\lambda_{mse}$, $\lambda_{ssim}$, and $\lambda_{lpips}$ are weights for each term. This loss term assesses the fidelity of the predicted output compared to the outputs of arbitrary ATM methods, incorporating Mean Squared Error (MSE), Structural Similarity Index Measure (SSIM) (61), and Learned Perceptual Image Patch Similarity (LPIPS) (62). This multifaceted approach ensures a comprehensive evaluation of reconstruction quality.

**Overall Loss.** The overall loss combines the similarity loss with temporal consistency and semantic enhancement:

$$\mathcal{L}_{total} = \mathcal{L}_{sim} + \lambda_{temp}\mathcal{L}_{temp}. \tag{5}$$

## 4 Experiments

### 4.1 Datasets and Training Details

We adopt several real-world datasets for evaluation, including the OTIS (63), HeatChamber (5), subset of BVI-CLEAR dataset (64), TSR-WGAN dataset (4) and DOST (65). We trained the ConVRT model individually on each video clip with a learning rate of $2 \times 10^{-3}$, using the Adam optimizer (66). For each video clip, the batch size equals to the number of frames in that clip. The spatial resolution of both the trainable spatial feature map and the canonical spatial feature map matches the original frame resolution after square cropping. The temporal resolution parameter $T_{\text{res}}$ was set to 5, with parameters $Q_1$ and $Q_2$ configured to 128 and 256, respectively. More details about the network settings are provided in the supplementary material. Training was conducted on a single RTX A6000.

### 4.2 Evaluation Strategy

We selected VRT(52), TMT(3), and DATUM(2) as the base video methods for ConVRT due to their state-of-the-art performance in video restoration and video ATM. TurbNet(5) is selected for base

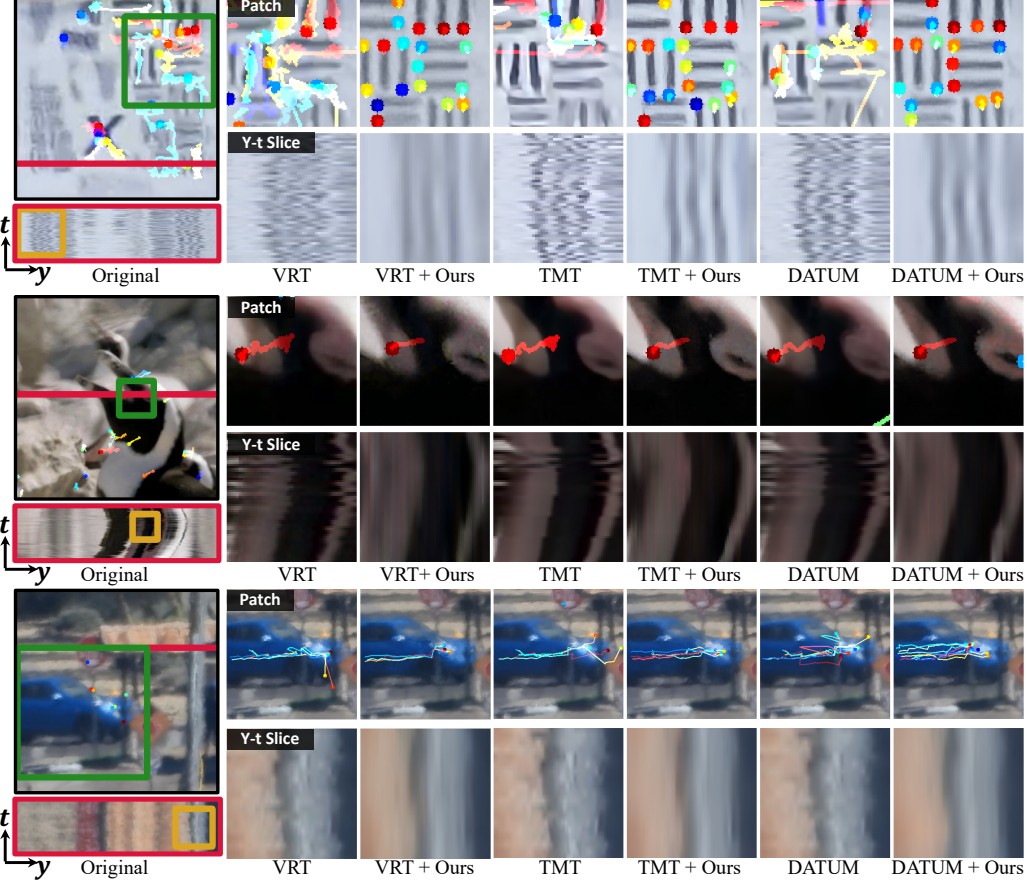

Figure 5: Visualization of our method's effectiveness in mitigating real-world atmospheric turbulence compared to existing methods. The leftmost image shows the original frame with a green box marking the zoom-in crop area for KLT tracking and a red line for the Y-t slice, shown in the bottom left. The right side displays two rows: the first shows zoom-in KLT tracking results for baseline methods and their outputs enhanced by our method, and the second shows zoom-in Y-t slices highlighting the temporal consistency achieved. Note the significant reduction in erratic movements in our results.

image method. We also directly applied ConVRT to the original video without the base methods to assess its standalone performance. To evaluate the consistency of turbulence removal in videos, we employed four metrics for quantitative evaluation and two interframe-related methods for qualitative assessment.

**Temporal Consistency and Per-frame Quality.** We used PSNR and SSIM to measure the per-frame reconstruction quality. Following (67), we utilized the average warp error to quantify the temporal consistency of the restored video. The warp error between two consecutive frames is defined as:

$$E_{\text{warp}}(V_t, V_{t+1}) = \frac{1}{\sum_{i=1}^{N} M_t^{(i)}} \sum_{i=1}^{N} M_t^{(i)} \left| V_t^{(i)} - \hat{V}_{t+1}^{(i)} \right|_2^2, \tag{6}$$

where $\hat{V}_{t+1}^{(i)}$ is the warped frame by optical flow at time $t + 1$ and $M_t^{(i)} \in {0, 1}$ is the occlusion mask estimated by the methods proposed in (68). The average warp error across the entire video sequence is calculated as:

$$E_{\text{warp}}(V) = \frac{1}{T-1} \sum_{t=1}^{T-1} E_{\text{warp}}(V_t, V_{t+1}). \tag{7}$$

| Turb Type | Dataset | Metrics | Video-based ATM | | | | | | Image-based ATM | | | No Base Method | | | Other Restoration Method | | |
|---|---|---|---|---|---|---|---|---|---|---|---|---|---|---|---|---|---|
| | | | TMT | + ConVRT | Gain | DATUM | + ConVRT | Gain | TurbNet | + ConVRT | Gain | Ori | + ConVRT | Gain | VRT | + ConVRT | Gain |
| Real | HeatChamber(63) | $E_{warp}$ ↓ | 24.21 | 19.78 | -4.43 | 22.43 | 17.77 | -4.66 | 41.48 | 17.36 | -24.12 | 24.33 | 16.19 | -8.14 | 25.77 | 16.19 | -9.58 |
| | | $PSNR_{x-t}$ ↑ | 18.45 | 18.60 | +0.15 | 19.41 | 19.60 | +0.19 | 18.40 | 19.33 | +0.93 | 23.86 | 24.18 | +0.32 | 23.84 | 24.18 | +0.34 |
| | | $Flow_{tv}$ ↓ | 5695.27 | 2786.99 | -2908.28 | 5794.12 | 2509.21 | -3284.91 | 17030.68 | 2383.29 | -14647.39 | 9471.83 | 2154.55 | -7317.28 | 9314.95 | 2154.55 | -7160.40 |
| | | PSNR ↑ | 18.41 | 18.59 | +0.18 | 19.25 | 19.46 | +0.21 | 18.27 | 18.98 | +0.71 | 19.79 | 19.96 | +0.17 | 19.69 | 19.96 | +0.27 |
| | | SSIM ↑ | 0.67 | 0.68 | +0.01 | 0.69 | 0.70 | +0.01 | 0.63 | 0.68 | +0.05 | 0.67 | 0.68 | +0.01 | 0.68 | 0.68 | +0.01 |
| | OTIS(63) | $Slice_{tv}$ ↓ | 1365.77 | 387.40 | -978.37 | 1237.91 | 365.09 | -872.82 | 3124.37 | 638.03 | -2486.34 | 1344.28 | 294.93 | -1049.35 | 1579.10 | 313.47 | -1265.63 |
| | | $Flow_{tv}$ ↓ | 7334.87 | 963.53 | -6371.34 | 6742.56 | 871.26 | -5871.30 | 11454.92 | 811.78 | -10643.15 | 7827.12 | 670.35 | -7156.77 | 8985.64 | 662.76 | -8322.87 |
| | CLEAR (64) | $Slice_{tv}$ ↓ | 115.34 | 109.71 | -5.63 | 129.34 | 113.92 | -15.42 | 377.62 | 210.78 | -166.84 | 172.82 | 104.76 | -68.06 | 186.76 | 105.31 | -81.45 |
| | | $Flow_{tv}$ ↓ | 3916.67 | 960.54 | -2956.13 | 4023.44 | 933.97 | -3089.47 | 11827.17 | 995.37 | -10831.80 | 8333.30 | 845.42 | -7487.88 | 9120.35 | 852.03 | -8268.32 |
| | TSRWGAN(4) | $Slice_{tv}$ ↓ | 129.22 | 135.70 | +6.48 | 123.32 | 124.93 | +1.61 | 523.47 | 311.07 | -212.40 | 151.65 | 115.07 | -36.58 | 168.90 | 118.22 | -50.68 |
| | | $Flow_{tv}$ ↓ | 2176.51 | 419.36 | -1757.15 | 2279.42 | 411.80 | -1867.61 | 6038.92 | 474.29 | -5564.63 | 3460.89 | 394.22 | -3066.67 | 3700.43 | 393.54 | -3306.89 |

Table 2: Performance improvements achieved by applying our proposed ConVRT across various model architectures and datasets. The No Base Method columns show the results when the methodology was applied directly to the original frames, labeled as Ori. Gains are highlighted for each metric, showing the effectiveness of ConVRT in enhancing the temporal consistency in video ATM.

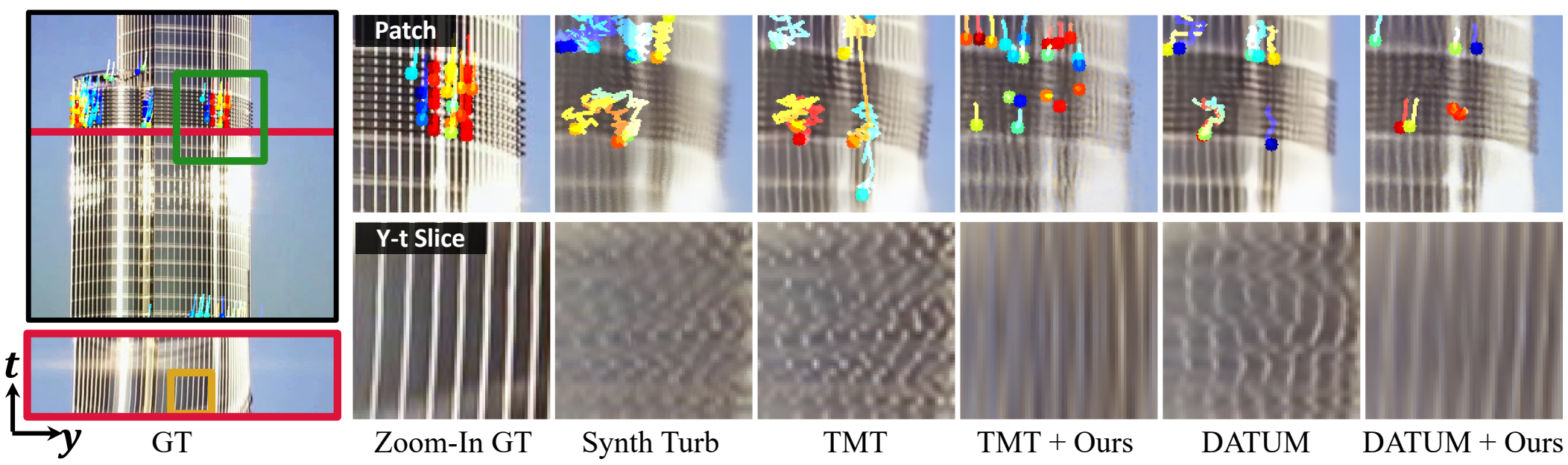

Figure 6: Comparison of turbulence mitigation techniques on a synthetic dataset. Left: Ground truth (GT) frame and corresponding Y-t slices. Right: Zoom-in views of KLT tracking (top row) and Y-t slices (bottom row) for baseline methods and the enhancements brought by our method.

In addition to the warp loss, inspired by (69), we also calculate the Total Variation loss of the X-t slice, $Slice_{tv}$, and Total Variation of the optical flow, $Flow_{tv}$, to quantitatively measure whether the temporal variation of time slices in restored videos are small.

**KLT Trajectories.** We employed the KLT tracker (70) to track feature points and plot their trajectories, as shown in Figure 5. KLT tracking is directly based on image gradient information, such that common issues in turbulence restoration, i.e., blurriness, artifacts, and temporal inconsistency, are reflected in the tracked trajectories. Smooth and coherent trajectories indicate temporally consistent restoration, while erratic or discontinuous trajectories suggest the presence of artifacts or inconsistencies.

**$x$-$t$ Slice.** We plotted $x$-$t$ slices to visualize the motion of a row of pixels, as illustrated in Figure 5. If the video restoration is temporally consistent, the $x$-$t$ slice plot will exhibit smooth and continuous curves. In contrast, non-smooth or jagged curves in the $x$-$t$ slice indicate temporal inconsistencies or artifacts in the restored video.

### 4.3 Qualitative and Quantitative Improvements on Existing Methods.

**Qualitative Real-world Cases.** Our method, ConVRT, achieves notable temporal consistency in videos distorted by real atmospheric turbulence. As shown in Figure 5, the original turbulence and baseline methods exhibit "zig-zag" KLT tracking trajectories, indicating erratic motion caused by turbulence. In contrast, incorporating ConVRT results in smoother trajectories, demonstrating its effectiveness in consistently removing turbulence artifacts throughout the video. The x-t slice further illustrates that ConVRT effectively smooths row pixel motion over time, reducing the flickering effects typi-

Table 3: Ablation Study of $L_{temp}$ and $T_{res}$. Comparison of $PSNR_{img}$, SSIM, and $PSNR_{x-t}$ scores, showing the impact of $L_{temp}$ and $T_{res}$. The experiment is conducted on a synthetic dataset created using turbulence simulator(15). The base model is TurbNet.

| Method | $T_{res}$ | $L_{temp}$ | $PSNR_{Img}$ ↑ | SSIM ↑ | $PSNR_{x-t}$ ↑ |
|---|---|---|---|---|---|
| TurbNet | - | - | 22.57 | 0.673 | 24.20 |
| + ConVRT | 15 | | 23.29 | 0.679 | 24.86 |
| + ConVRT | 8 | | 23.91 | 0.694 | 25.51 |
| + ConVRT | 5 | | 24.16 | 0.701 | 26.02 |
| + ConVRT | 5 | ✓ | **24.31** | **0.709** | **26.05** |

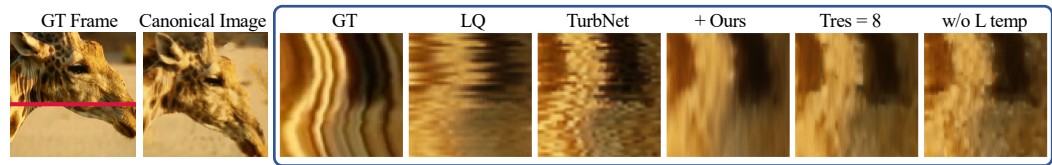

Zoomed Y-t Slice

Figure 7: Ablation study and canonical image visualization. Our method mitigates residual turbulence using $L_{\text{temp}}$ and lower $T_{\text{res}}$. Canonical image is visualizvisualized from Canonical Spatial Feature Map C.

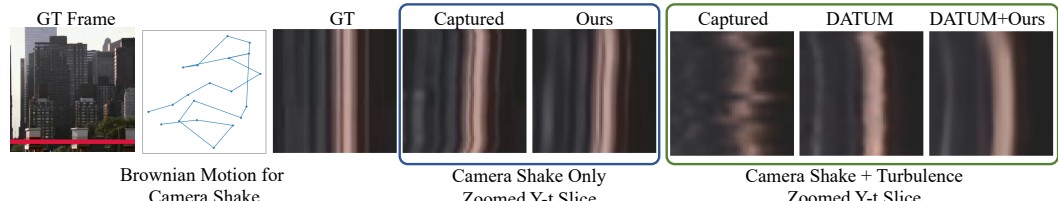

Figure 8: Illustration of camera shake simulation using Brownian Motion. In the Y-t slice plots, we observe similarities between camera shake and turbulence. The plots also demonstrate the effectiveness of our approach in handling both camera shake alone and in combination with turbulence.

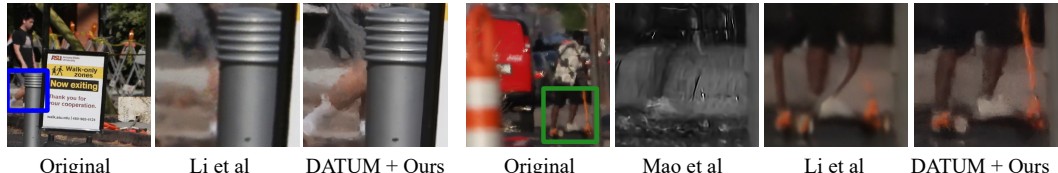

Figure 9: Experimental results compare ConVRT with unsupervised and test-time optimization methods by Li et al. (10) and Mao et al. (9) on moving objects. Both baselines fail to capture motion, replacing moving parts with the average background, while ConVRT effectively handles them

cally observed in turbulence-distorted videos. This improved performance underscores the capability of ConVRT to handle real-world turbulence, providing more stable and visually coherent video sequences.

**Qualitative Synthetic Cases.** Similarly, on synthetic video, As shown in Figure 6, our method enhances temporal turbulence removal when applied to baseline methods. The video dynamics generated by ConVRT closely resemble the ground-truth videos, effectively smoothing out atmospheric turbulence. The improved KLT trajectories further demonstrate this temporal consistency.

**Quantitative Results.** We evaluated the performance of our proposed ConVRT method across real-world datasets containing both static and dynamic scenes, as shown in Table 2. ConVRT demonstrates consistent improvements across models and most of datasets, underscoring its broad applicability. On the HeatChamber dataset, which provides real-world paired data through a controlled heating mechanism, we calculated PSNR values to further substantiate ConVRT's effectiveness. onVRT consistently improves PSNR, demonstrating robust enhancement of temporal consistency, especially given PSNR's sensitivity to pixel misalignment.

## 4.4  Ablation Study

Regularized temporal resolution $T_{\text{res}}$ is critical for ensuring temporal consistency. Lowering it results in smoother transitions but loses fine details, while a higher value preserves details but increases the risk of flickering. We conducted an ablation study on the impact of $T_{\text{res}}$ and $L_{\text{temp}}$, as shown in Table 3, with qualitative results in Figure 7. These results demonstrate the effectiveness of our representation field design in regularizing irregular turbulence motion.

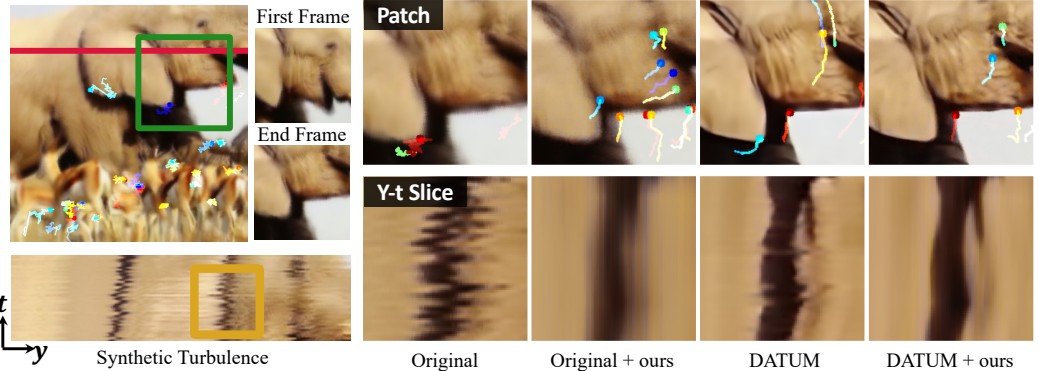

Figure 10: Mitigation capability of our method without using base restoration methods for pre-processing. The scene is an elephant raising its head.

### 4.5 Analysis

**Why It Works.** Our method's effectiveness stems from two key factors. First, it leverages the distinct differences in motion patterns, particularly the regularity of optical flow directions within a short time window, between regular object movement and atmospheric turbulence, as discussed in Section 1.2. This distinction also enables our method to handle camera shake, which shares similar irregular patterns with turbulence. The results of this capability are illustrated in Figure 8. Second, our method includes a robust video representation that overcomes limitations in unsupervised methods for static scenes. As illustrated in Figure 9, these methods often struggle on moving objects, blending them into static backgrounds. In contrast, our approach preserves object integrity across frames, making it well-suited for video-based turbulence mitigation.

**Mitigation Capability without Base Restoration Methods.** Even without a base restoration method to provide partially restored frames, our approach could improve temporal consistency, as shown in Figure 10. However, we recommend combining our method with other restoration techniques. This allows user to benefit from the sharpness improvements offered by supervised methods, while also taking advantage of the temporal consistency improvements provided by ConVRT.

**Visualizing Trainable Feature Maps.** We visualize the canonical image by inputting the canonical spatial feature map into the content MLP without applying $\Delta x$ and $\Delta y$, as shown in the Figure 7. The canonical image contains most of the video's content, providing a base representation from which other frames can be derived. Consequently, the canonical spatial field in our video representation functions similarly to a key frame in video compression, serving as a reference for other frames in the sequence to query information.

## 5   Limitations

While ConVRT offers significant improvements in video atmospheric turbulence mitigation, there are two limitations. First, as a neural representation method, ConVRT's performance depends on accurate video representation and currently optimized to capture motion with precision in short clips. Extending this to longer sequences and more complex motions is a potential area for future exploration. Second, ConVRT processes a 25-frame video at 540x540 resolution in approximately 10 minutes, including DATUM as base method. Although much faster than Mao's (165 minutes) and Li's (300 minutes) methods, there is still room for improving computational efficiency, especially for larger or more complex sequences.

## 6   Conclusion

In this paper, we present ConVRT, a novel approach aimed at enhancing temporal consistency in video ATM tasks. ConVRT uses a dual-field approach—Temporal Deformation Field and Spatial Content Field—to accurately capture spatial information while regularizing temporal information, focusing on regular object motion rather than irregular turbulence. Combined with any ATM method, ConVRT leads to visibly improved temporal consistency.

**Acknowledgment.** H.C., M.X., K.Z., and C.A.M. were was supported in part by AFOSR Young Investigator Program Award no. FA9550-22-1-0208, ONR award no. N000142312752, and NSF CAREER Award no. 2339616. W.J. and A.V. were supported in part by ONR award no. N00014-23-1-2714.

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

## A  Appendix / supplemental material

### A.1  MLP Network Details

Our network architecture consists of two MLPs: a content MLP and a deformation MLP. The content MLP has 4 fully-connected layers, with an input dimension of $Q_1$, a hidden dimension of 128, and an output dimension of 2, representing $\Delta x$ and $\Delta y$. The deformation MLP comprises 6 fully-connected layers, with an input dimension of $Q_2$, a hidden dimension of 256, and an output of 3 channels representing RGB intensity.

### A.2  Position Encoding

Position encoding for spatial and temporal indices is embedded within the trainable feature map, as these indices are trainable. We directly use $x$, $y$, and $t$ to query the corresponding feature tensors from the feature maps. Notably, in the temporal feature map, neighboring features are shared across multiple frames, with each frame weighted differently due to explicit regularization.

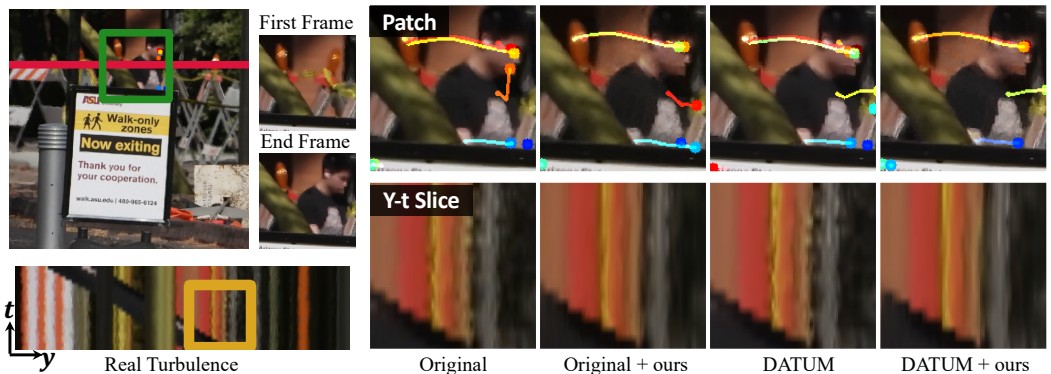

Figure 11: Mitigation capability of our method without using base restoration methods for pre-processing.

## A.3 Additional Results on Mitigation Capability without Base Restoration Methods.

Additional results highlighting our method's mitigation capability independently of base restoration techniques are presented in Figure 11.

