# OpenReview forum: "Temporally Consistent Atmospheric Turbulence Mitigation with Neural Representations"
_NeurIPS.cc/2024/Conference — NeurIPS 2024 poster_

### Official Review · Reviewer_VLLH · 2024-06-17

**Soundness:** 3
**Presentation:** 3
**Contribution:** 3
**Rating:** 4
**Confidence:** 5

**Summary:**

The study focuses on Video Atmospheric Turbulence Mitigation (ATM), which aims to restore videos that are affected by distortions caused by atmospheric turbulence. Specifically, the proposed ConVRT introduces a neural video representation that decouples spatial and temporal information, allowing targeted regularization of the network's temporal representation capability. Also, this paper integrates supervised and self-supervised learning, significantly improving the temporally consistent mitigation of ATM methods on diverse real-world data.

**Strengths:**

The study introduces a novel framework called ConVRT for video ATM, which addresses the challenge of maintaining temporal consistency in turbulence mitigation. It proposes a neural video representation that decouples spatial and temporal information, allowing targeted regularization and effective mitigation of turbulence-induced temporal frequency variations. Furthermore, the study integrates supervised and self-supervised learning, improving the temporally consistent mitigation of ATM methods on diverse real-world data.

**Weaknesses:**

1.	As a neural representation method, ConVRT is designed to handle video clips with a limited number of frames, making it challenging to handle larger video sequences and more significant motions without compromising accuracy.
2.	The related work section lacks completeness. More test-time methods should be carefully reviewed. Also, no SOTA methods were involved in the experimental section for comparison, which is unacceptable.
3.	In addition to VRT, more SOTA transformer-based methods, as well as the recent-popular diffusion-based models, should be added for analysis.
4.	ConVRT employs test-time optimization, which can be computationally intensive. Therefore, model efficiency should be discussed.
5.	Some intermediate results should be given.

**Questions:**

Please refer to the detailed comments!

**Limitations:**

Please refer to the detailed comments!

---

> ### Author Rebuttal · Authors · 2024-08-07
>
> **`R4-Q1`**: **Capability of handling longer video sequences and significant motion**.
>
> Our method effectively handles videos with significant motion. We've included additional experimental results and a detailed analysis in our global response to **`shared question A`**. This demonstrates our approach's robustness across various dynamic scenarios
>
> **`R4-Q2`**: **The related work section lacks completeness. More test-time methods should be carefully reviewed. Also, no SOTA methods were involved in the experimental section for comparison, which is unacceptable.**.
>
> We'd like to highlight that our experimental section includes a comprehensive setup that SOTA and advanced methods in the related fields:
>
> * TMT (TCI’23) [1]: The pioneering video-based ATM method utilizing transformer architecture and temporal attention.
>
> * DATUM (CVPR’24)  [2]: The current SOTA video-based ATM method, trained on an advanced turbulence simulator dataset, achieving 10x faster inference time and superior performance compared to TMT.
>
> * TurbNet (ECCV’22) [3]: An effective image-based method trained on an advanced turbulence simulator.
>
> * VRT (TIP’24) [4]: A notable video method for general deblurring.
>
> Comprehensive qualitative and quantitative results are presented in  Figure 1, Figure 5 and Table 1 of the main paper, with additional comparisons shown in **`attached file Figure 1`**. These results demonstrate our method's significant performance improvements in addressing the residual distortion (temporal inconsistency) that persists in current SOTA methods.
>
> Table 1 and subsection 1.1 in the main paper present a comprehensive taxonomy of current ATM methods, analyzing their strengths and limitations relative to our approach.
>
> We've added two more unsupervised and test-time optimization-based turbulence removal methods for comparison and discussion in our global response to **`shared question C`**.
>
> [1] Zhang, Xingguang, et al. "Imaging through the atmosphere using turbulence mitigation transformer." IEEE Transactions on Computational Imaging (2024).
>
> [2] Zhang, Xingguang, et al. "Spatio-Temporal Turbulence Mitigation: A Translational Perspective." Proceedings of the IEEE/CVF Conference on Computer Vision and Pattern Recognition. 2024.
>
> [3] Mao, Zhiyuan, et al. "Single frame atmospheric turbulence mitigation: A benchmark study and a new physics-inspired transformer model." European Conference on Computer Vision. Cham: Springer Nature Switzerland, 2022.
>
> [4] Liang, Jingyun, et al. "Vrt: A video restoration transformer." IEEE Transactions on Image Processing (2024).
>
>
>
> **`R4-Q3`** : **In addition to VRT, more SOTA transformer-based methods, as well as the recent-popular diffusion-based models, should be added for analysis**.
>
> Regarding the comparison with diffusion-based models, we've prepared a comprehensive analysis highlighting our method's unique advantages. Please see our response to **`R2-Q5`** for an in-depth discussion on this topic.
>
> **`R4-Q4`**: **Discussion about Efficiency**.
>
> We appreciate your suggestion on our model's efficiency. Our global response to **`R1-Q5`** provides a detailed breakdown of our method's computational performance, including comparisons with other test-time optimization approaches for turbulence removal.
>
> **`R4-Q5`**: **Intermediate results**.
>
> These results are compiled in our global response to **`shared question D`** for a clearer understanding of our approach.

---

> > ### Comment · Reviewer_VLLH · 2024-08-11
> > **Response to Author Rebuttal**
> >
> > I appreciate the author's rebuttal.  However, I am still concerned about the capability for large motion and moving objects.
> >
> > Therefore, I have decided to maintain my original rating of "Borderline reject".

---

> > > ### Author Response · Authors · 2024-08-12
> > > **Follow-up Request**
> > >
> > > Would you mind elaborating on your concern?
> > > As illustrated in Figure 1 of the response, our approach can effectively reconstruct scenes with substantial motion: Over the course of the video the person moves across nearly half the field-of-view (see the long diagonal band in the x-t slice).

---

### Official Review · Reviewer_7Q3b · 2024-07-14

**Soundness:** 3
**Presentation:** 3
**Contribution:** 3
**Rating:** 5
**Confidence:** 4

**Summary:**

This paper proposes a method for improving the temporal consistency of turbulence-affected videos. The proposed method uses neural representations (MLP layers) to separately model the spatial and temporal deformations caused by air turbulence and is able to improve the temporal consistency of restoration results. It seems that the proposed method needs to be used in conjunction with another base turbulence restoration method. The prosed method (by combining with several other SOTA approaches) is evaluated on existing real turbulent video dataset, and a small dataset collected by the authors. Temporal consistency, especially for videos with moving objects, is apparently improved after applying the proposed method.

**Strengths:**

- The paper is generally well-written and easy to follow.
- The MLP-based network for mitigating spatial and temporal deformations is new and seems to be effective, especially on maintaining temporal consistency.
- Qualitative evaluation is performed on videos with moving objects and have shown apparent improvement.
- A small real-world dataset was captured and used for testing, but details about the dataset are not clear.

**Weaknesses:**

- It seems that the method is an add-on approach for regularizing the temporal consistency of videos restored by another turbulence restoration methods. Since turbulent degradation is more complex than temporal inconsistency and spatial distortions (e.g., there might be blurriness and color aberration beyond the spatial and temporal deformations), being able to only handle these two types of artifacts seems quite limited for turbulence mitigation.
- The quantitative evaluation results (Table 2) are confusing. This table shows metric scores for using the proposed method as a stand-alone (i.e., using the original turbulent video as input). However, the "no base results" are not demonstrated in any visual comparison figures (not even in supplementary materials). It would be useful to see visual comparison between "no base results" and others. What really confuses me about this table is that the metric scores of the original turbulent images (denoted as "ori") are even better than processed results in many cases (for example, its PSNR is higher than DATUM results for the HeatChamber dataset, and there many such cases). But according to visual results, most processed results have apparent improvement. Besides, after applying the proposed method, some metric scores become much worse (for example, the slice_tv scores for most datasets and method combinations). There should be some discussions explaining the metric results.
- The paper missed some relevant prior works (see below). These works either use MLP for modeling turbulent distortions or use similar idea to enforce temporal consistency and they should be discussed and compared with the proposed method.
Li et al., "Unsupervised Non-Rigid Image Distortion Removal via Grid Deformation," ICCV 2021.
Thapa et al. "Learning to Remove Refractive Distortions from Underwater Images," ICCV 2021.

**Questions:**

- It seems that motions in demonstrated results are quite slight and slow. Is the method also robust to large motions? How is this related to turbulence strength?
- I'm interested at knowing more details about the airport dataset acquired by the authors, like size of the dataset, type of scenes, turbulence strength, etc.

**Limitations:**

Two limitations are discussed: 1. length of input video cannot be too long, and 2. processing time is sort of long (running time of the method is not reported). Turbulence strength and motion scale could be also discussed, if they are limiting factors.

---

> ### Author Rebuttal · Authors · 2024-08-07
>
> **`R3-weakness 1`**:  **Forward Model of Turbulence.**
>
> Indeed mitigating turbulence requires overcoming color aberrations, blurriness, and deformations and other effects. Fortunately, existing techniques have successfully addressed many of these challenges and arguably the largest limitation of existing approaches is that their results lack temporal consistency. Our method addresses this challenge.
>
> Recent advancements in the field, from TMT (2023) to DATUM (2024), have leveraged increasingly large datasets created by advanced simulators and more complex networks. While these improvements have significantly enhanced sharpness in real-world cases, as evidenced in **` main paper Figures 1`** and **`main paper Figure 5`** of our main paper, poor temporal consistency remains a glaring issue. This inconsistency is not only noticeable to the human eye but also detrimental to downstream tasks such as video segmentation.
>
> This persistent problem suggests that video-based ATM methods struggle to improve temporal consistency relative to other types of degradation. The primary challenge lies in poor generalization to the random and complex deformations caused by real-world turbulence.
>
> Our method addresses this critical gap by providing an effective refinement solution for video-based ATM methods when applied to real-world videos. We believe that tackling the temporal inconsistency issue represents a significant contribution to the video ATM field at this stage, as it targets a key weakness in current state-of-the-art approaches
>
>
> **`R3-weakness 2`**: **Qualitative result of ours trained on the distorted video (No Base Method).**
>
> Even without a base method to provide partially restored frames, our approach effectively improves temporal consistency by a significant margin, as demonstrated in **` attached file Figure 1`**. For clear evidence of this improvement, please observe the X-t slice presented there.
>
>
> **`R3-weakness 2`**: **Table 2 is confusing. Why do video-based ATM methods have worse PSNR and PSNR_x-t compared to image-based methods on the HeatChamber dataset?**
>
> Apologies for any confusion caused by the content in the main table. Below is a detailed explanation of our results. We will clarify in the revision.
>
> When reconstructing static scenes, image-based reconstruction methods have a significant advantage over video-based methods. This is because image-based methods effectively impose a prior assumption that the scene is completely static, while video-based methods do not. Consequently, it's not surprising that image-based methods perform well on the static HeatChamber dataset.
>
> Without this strong static prior, it's understandable that video-based methods are less effective on static datasets. However, this situation reverses for dynamic scenes, where the static scene prior would be invalid. In these cases, image-based methods perform much worse than video-based methods.
>
> **`R3-weakness 2`**: **Why do some metric scores (tv_slice) appear much worse?**
>
> This is a typo; a lower slice_tv indicates better performance. We will fix it in the camera-ready version.
>
> **`R3-weakness 3`**: **Missings comparisons two unsupervised methods**
>
> At present, no open source implementation of [3] is available. We have provided comparisons with Mao’s [1] Li's [2] in our global response to the **` shared question C`**. Our approach handles videos with moving objects, whereas Li's and Mao's methods cannot.
>
> [1] Mao, Zhiyuan, Nicholas Chimitt, and Stanley H. Chan. "Image reconstruction of static and dynamic scenes through anisoplanatic turbulence." IEEE Transactions on Computational Imaging 6 (2020): 1415-1428.
>
> [2] Li, Nianyi, et al. "Unsupervised non-rigid image distortion removal via grid deformation." Proceedings of the IEEE/CVF International Conference on Computer Vision. 2021.
>
> [3] Thapa, Simron, Nianyi Li, and Jinwei Ye. "Learning to remove refractive distortions from underwater images." Proceedings of the IEEE/CVF International Conference on Computer Vision. 2021.
>
>
> **`R3-Q1`** : **Capability on large motion**.
>
> Yes, our method can handle moving objects with large motion. For the qualitative results, please see our global response to the **` shared question A `** .

---

> > ### Comment · Reviewer_7Q3b · 2024-08-13
> >
> > My concerns are mostly addressed by the rebuttal. I would support the acceptance of this paper, given its strength on handling moving objects and improving the temporal consistency of turbulence restoration. In the final version, the author should clarify quantitative results and include comparisons with the methods by Li et al. and Mao et al.

---

> ### Author Response · Authors · 2024-08-14
>
> Thank you for your thoughtful review and for recognizing the potential of our paper.
>
> In the revised version, we will include both quantitative and qualitative comparisons with unsupervised and test-time optimization methods ( Li et al. & Mao et al). Additionally, we will expand our discussion section to incorporate your insightful questions and our exchange, helping readers better understand our contribution in the atmospheric turbulence mitigation field.
>
> Again, we greatly appreciate your valuable time and insightful suggestions.

---

### Official Review · Reviewer_EL8T · 2024-07-26

**Soundness:** 2
**Presentation:** 2
**Contribution:** 2
**Rating:** 5
**Confidence:** 4

**Summary:**

This paper introduced ConVRT, a novel method for video atmospheric turbulence mitigation.
This paper has a good structure and is well-written.

**Strengths:**

This paper proposed a new method to deal with turbulence mitigation.

**Weaknesses:**

1. limited real-world case visualization
2. limited proof of algorithm effectiveness
3. limited comparison with classic algorithm

**Questions:**

1. Test cases with visualization are so limited, even in the supplemental material. Please show more real-world cases to show the effectiveness of the algorithm.
2. As the representative unsupervised method, "Unsupervised non-rigid image distortion removal via grid deformation," it is important to compare with it, no matter from the algorithm or qualitative result.
3. Lack of ablation studies to prove the effectiveness of the proposed algorithm.
4. Can the method deal with the video with moving objects?
5. No matter whether it is for a single image-based or multiple frames-based turbulence mitigation, most existing algorithms can deal with them very well. With the help of the diffusion model, the resulting image can be refined further. It means that it could generate a good final result in a short time.  Then, what is the advantage of your algorithm?
6. It is important to visualize the temporal and spatial field to verify the algorithm's effectiveness.

**Limitations:**

Same with question

---

> ### Author Rebuttal · Authors · 2024-08-07
>
> **`R2-Q1 & R2-Q4`**: **Capability of moving object on more cases**
>
> Yes, our method can handle moving objects. More than half of the cases in our main paper and supplementary materials are dynamic videos. This is evident as the lines in X-t Slice or Y-t Slice are not perfectly vertical or horizontal, indicating the objects are moving along the temporal dimension. Please also view the html file in the supplement which includes video reconstructions of our scenes.
>
> We have also added additional experimental results, with large object motion, in our global response to **`shared question A`**. Our approach can effectively reconstruct scenes with significant object motion.
>
> **`R2-Q2`**:  **Comparison with Li’s method**
>
> We have added comparisons with Li's [1] in our global response to **`shared question C`**.  Our approach handles videos with moving objects, whereas Li's method cannot.
>
> [1] Li, Nianyi, et al. "Unsupervised non-rigid image distortion removal via grid deformation." Proceedings of the IEEE/CVF International Conference on Computer Vision. 2021.
>
> **`R2-Q3`**: **More ablation study.**
>
> We conducted an ablation study on temporal resolution (T) and the presence of L_temp, as shown in **`Table below`**, with corresponding qualitative results in **`attached file Figure 4`**.
>
> The first row of the Table shows TurbNet's performance. Equipping TurbNet with our method yields a 0.76 dB improvement in PSNR, demonstrating the effectiveness of our representation field design in regularizing irregular turbulence motion. Decreasing T from 15 to 5 (implying stronger regularization) further improves PSNR by 0.94 dB, SSIM by 0.03, and PSNR_slicext by 1.28 dB. Adding our proposed loss function achieves additional performance gains.
>
> **`Attached file Figure 4`** illustrates qualitative results. Removing the proposed loss function and relaxing regularization leads to noisier and more jittery X-t slices compared to our full method
>
>
> | Method Name | Resolution of T| Our Feature Maps  | L_temp | PSNR | SSIM | PSNRx-t |
> |--------------|:------------------:|:------------------:|:------------------:|:------------------:|:------------------:|:------------------:|
> | TurbNet      |                    |               |        |    23.46  |   0.68   |   26.08 |
> | Ours            | 15               | ✔️            |        |    24.22  |   0.69   |   26.63 |
> | Ours            | 8               | ✔️            |        |    24.88  |  0.70    |   27.46 |
> | Ours            | 5               | ✔️            |        |   25.16   |   0.72   |   27.91 |
> | Ours            | 5               | ✔️             | ✔️      |  25.29    |   0.73   |   27.99 |
>
>
> **`R2-Q5`**: **Advantage of the proposed method over diffusion model.**
>
> Our method in video ATM offers several advantages over diffusion models: It is generalizable, physics-grounded, temporally consistent, and free from hallucinations.
>
> Diffusion models can learn incredibly accurate priors on images and, more recently, videos. Long term, they could be an effective method to mitigate turbulence. However, they also come with severe limitations. Diffusion models require extensive training data and often suffer from hallucination biases inherent in that data. These models are purely data-driven and do not consider domain knowledge or the physics of turbulence. In contrast, our method addresses residual temporal inconsistency issues from existing physics-based turbulence removal models. By observing turbulence motion features, we design the mitigation based on temporal motion regularity, achieving effective results without any training. Moreover, our method can be applied to any existing physics-based ATM method, making it more grounded in the physics of turbulence rather than relying solely on training data quality.
>
> To date, there has been no exploration of diffusion models for video-based ATM. The only diffusion model method, PiRN, for image-based ATM is a closed-source approach trained on limited synthetic image datasets. There is no evidence it can reconstruct temporally consistent dynamic videos. In contrast, our method significantly improves temporal consistency.
>
> **`R2-Q6`**: **Visualization of the temporal and spatial field.**
>
> Please see our global response to the **`shared question D`**

---

> > ### Comment · Reviewer_EL8T · 2024-08-11
> > **Solve most of my conserns**
> >
> > For 'Advantage of the proposed method over diffusion model', I mean that it could be an further improve the result of some algorithms turbulence mitigation result.

---

> > > ### Author Response · Authors · 2024-08-12
> > > **Further discussion with Reviewer EL8T (denoted as R2)**
> > >
> > > Thank you for your further explanation.
> > >
> > > Applying diffusion models as add-ons to ATM methods presents challenges with hallucination and may worsen temporal consistency. Current restoration-focused diffusion models still struggle with hallucination. Using these models on ATM video outputs has a high likelihood of exacerbating temporal inconsistency, as turbulence-induced random local motion in each frame may result in inconsistent texture restoration, potentially degrading video quality. We will include a small figure in our revised version to further illustrate the limitations of the diffusion model as adds-on.
> > >
> > > However, your suggestion points to a valuable but unexplored direction for future ATM research. To effectively use diffusion models as add-ons to refine current ATM methods, future work might consider developing a specialized turbulence module within video-based diffusion models. Our work could inspire such developments, potentially including a regularized temporal attention module.
> > >
> > > We appreciate your response. Please let us know if you have any further questions or concerns. We'll respond promptly to your comments.

---

> > > > ### Comment · Reviewer_EL8T · 2024-08-12
> > > >
> > > > I have no more questions. Thanks

---

> > > > > ### Author Response · Authors · 2024-08-14
> > > > >
> > > > > Thank you for letting us know our rebuttal addresses all your concerns. We will incorporate all your suggestions in the camera-ready version (if it is accepted), to help readers better understand our method and its contribution to the ATM field. Thank you again for your valuable time and comments

---

### Official Review · Reviewer_DNNj · 2024-07-27

**Soundness:** 3
**Presentation:** 3
**Contribution:** 3
**Rating:** 6
**Confidence:** 4

**Summary:**

This paper presents an implicit neural representation (INR) framework for taking a pre-trained supervised video atmospheric turbulence mitigation (ATM) model and regularizing its output to be more temporally consistent. The main components are (1) an INR called the temporal deformation field; and (2) a subsequent INR called the spatial content field to output RGB intensity at a pixel in the video (at a certain time). These two INRs are trained on the output of a pre-trained ATM model, and regularized using a disparity loss (with MiDas pre-trained network) for temporal consistency, and a similarity loss for the content of the video. Experiments are conducted on real-world datasets with comparison to state-of-the-art ATM models recently proposed as well as some simulated ablation studies.

**Strengths:**

+ Method can improve a variety of existing state-of-the-art ATM models, and the use of INRs with different feature representations used as inputs seems like an original contribution (at least to this application field, if not video representation in general)

+ The use of KLT tracker for visualizing the temporal variability across the frames is a good visualization and helps show the improvement of the method in a qualitative way

+ Supplemental videos on the website show the method stabilizing video in the presence of turbulence

+ Extensive quantification of the method shown in Table 2 to illustrate the effectiveness of ConVRT

**Weaknesses:**

- There is little detail about the spatial feature map M, temporal feature map N, canonical spatial feature map C. What does it mean to call these feature maps, and why are they chosen the way they are? For instance, why the Hadamard product for M and N, and not just learning a 3D feature map at that place instead directly? I also don't see how the C map is "canonical" to me in any obvious way (for instance, you could change the dimensions of Q1, Q2 and I don't see why that couldn't work in the method?).

- The method seems to be focused primarily on fixing errors for supervised ATM methods. However, some of the classical approaches such as [Mao 2020] that utilize lucky frames, would they have this problem of temporal variability? I'm not necessarily asking for a comparison or new experiments, but it would be good to discuss if this problem primarily is for supervised methods.

Reference: Zhiyuan Mao, Nicholas Chimitt, and Stanley H. Chan, ‘‘Image Reconstruction of Static and Dynamic Scenes through Anisoplanatic Turbulence’’, IEEE Transactions on Computational Imaging, vol. 6, pp. 1415-1428, Oct. 2020

- Table 3 is a very modest improvement . Does the Ltemp really help? A qualitative example would really help clear up that this L_temp is working (show a figure with and without Ltemp).

- How would the method handle issues such as camera shake (common in long-range videos that are shot with high optical zoom)?

Minor suggestions:
- Line 187 - TurbNet, shouldn't it be whatever ATM method you are comparing with?
- Line 205 - One shouldn't be capitalized
- Line 109 - unresolved citation
- Table 1 - more stylistic, but I don't think its necessary to put down the venue into the table. We shouldn't judge methods based on their venue, but on the content of the method itself, so having the citation alone is enough to let readers draw their own conclusions about the papers. I would remove this column from the table.
- Table 3 - there is a typo in the PSNR_Img column where the lower number is bolded rather than the higher one

**Questions:**

1. Can the authors explain (1) how the feature maps M, N, C are novel compared to other INR video representations? Is it the use of the Hadamard product, and the two stage architecture? If so, an ablation study of comparing the Hadamard product of M and N to just a 3D spatial feature map directly is warranted. (2) I would be good to visualize these features after optimization (what do they look like), and what information/interpretability can be gleaned from them?

2. There are missing details: what is the details of the MLP layers, any positional encoding? These should be added to a supplemental file.

3. I am interested if L_temp is the key factor that improves the method, or its a minor improvement. Showing a qualitative example as discussed earlier would be beneficial here.

4. Can there be a discussion about issues involving camera shake? I assume the method would require stabilized videos first, and if there is any residual motion leftover, this would cause errors in the reconstruction.

5. What is the wall clock time of the method? How long does it take (in actual seconds) from start to finish? The paper only states 80 epochs, I'm curious how long that actually takes.

6. In line 134 - it says an additional enhancement module is applied after S_field. This was never discussed again. How important is this module? What's the performance with and without the module?

**Limitations:**

Yes

---

> ### Author Rebuttal · Authors · 2024-08-07
>
> **`R1-Q1 / R1 - weakness 1`** : **Visualization of Canonical Spatial Field and Representation Field Design**
>
> The canonical spatial field C serves as a base spatial representation, containing all spatial content of the video. We can obtain a canonical image by deriving it from this field without applying delta x and y. This canonical spatial field functions similarly to a key frame in video compression, providing a reference from which other frames in the sequence can be derived. For more detailed results and discussion on this topic, please refer to our global response to **`shared question D`**
>
> Our feature maps M and N, combined with the Hadamard product, provide an efficient representation strategy for 3D feature volumes represented by INR. This tensor decomposition strategy is commonly used to parameterize 3D volumes in INR, enhancing their ability to represent 3D signals while reducing the number of required parameters [1][2][3]. The Hadamard product in our two-stage architecture efficiently combines spatial and temporal information, offering a balance between computational efficiency and representational power, particularly suited for video ATM tasks.
>
>
> **`R1-Q3 / R1 - weakness 3`**: **More ablation study on loss function**
>
> Both representation and loss design are effective in our method. The neural representation design plays a more crucial role, bringing a 0.95dB improvement in performance, while the L_temp loss contributes an additional 0.15dB improvement, as demonstrated in our ablation study. For more details, please refer to our response to **`R2-Q3`**.
>
>
> **`R1-Q4 / R1-weakness 4`** : **Camera shake**
>
> Thank you for the interesting question. Please see our global response to **`shared question B`**.
>
>
> **`R1-Q5`**: **Wall clock time**
>
> We conducted experiments on a 25-frame video with 540x540 resolution. Our total running time consists of two parts: the preprocessing using the ATM model (DATUM) and our test-time optimization approach. For comparison, the running time for other unsupervised test-time optimization methods (Li's [2] and Mao’s [1] methods) only includes the methods themselves, as they do not rely on preprocessing.
> As shown in the **`Table`** below, even with the additional DATUM preprocessing time, our method is more than 15 times faster than Mao’s method and around 30 times faster than Li's. Our method's efficiency stems from combining the advantages of supervised models with test-time optimization. We use faster supervised methods to address major issues in video turbulence mitigation and apply the slower test-time optimization only to residual temporal inconsistencies. This approach makes our method significantly more efficient than pure test-time optimization methods that must solve the entire problem independently.
>
>
> | Method             | Running Time (min) |
> | :----------------: | :------:     |
> | Mao                |  165         |
> | Li               |  300         |
> | DATUM + Ours         |   10         |
>
>
> **`R1-Q2`**: **Missing details about MLP**
>
> Our network architecture consists of two MLPs: a content MLP with 3 fully-connected layers (input: Q1, hidden: 128, output: 2) and a temporal MLP with 8 fully-connected layers (input: Q2, hidden: 128, output: 3 RGB). We encode position information in the optimizable Canonical Spatial Feature Map C and Spatial Feature Map M, both with dimensions H x W matching the image. For each RGB image location, we query the corresponding Spatial Feature Map M location, perform a Hadamard product, and input to the MLP. In the Temporal Feature Map, multiple frames share neighboring features with different weights due to explicit regularization. We will provide additional illustrations in the revision to further clarify these concepts.
>
> **`R1-Q6`**: **The missing enhanced module**
>
> The “enhanced module” refers to a network feature that ablation studies caused us to remove. Referencing it in the manuscript was a typo—apologies for the confusion. This reference will be removed.
>
>
> **`R1 - weakness 2`**: **Add comparison with Mao’s method**
>
> Our method substantially outperforms Mao et al. [4] and [5] on dynamic video content. This is because their approaches are only effective for static scene turbulence removal and fail when applied to videos containing moving objects. For qualitative comparison and discussion, please see our global response to **`shared question C`**
>
> **`R1 - weakness 6`**: **Term involved in the loss function**
>
> At line 187, the loss is calculated between the output of our method and the output of the base method (TurbNet, TMT, DATUM, etc.). We will clarify this description in the camera-ready version.
>
> **`R1 - weakness 5/7/8/9`**: **Typos**
>
> Thank you for pointing out the typos. We will fix them in the camera-ready version.
>
>
> [1] Chen, Anpei, et al. "Tensorf: Tensorial radiance fields." European conference on computer vision. Cham: Springer Nature Switzerland, 2022.
>
> [2] Fridovich-Keil, Sara, et al. "K-planes: Explicit radiance fields in space, time, and appearance." Proceedings of the IEEE/CVF Conference on Computer Vision and Pattern Recognition. 2023.
>
> [3] Zhou, Haowen, et al. "Fourier ptychographic microscopy image stack reconstruction using implicit neural representations." Optica 10.12 (2023): 1679-1687.
>
> [4] Mao, Zhiyuan, Nicholas Chimitt, and Stanley H. Chan. "Image reconstruction of static and dynamic scenes through anisoplanatic turbulence." IEEE Transactions on Computational Imaging 6 (2020): 1415-1428.
>
> [5] Li, Nianyi, et al. "Unsupervised non-rigid image distortion removal via grid deformation." Proceedings of the IEEE/CVF International Conference on Computer Vision. 2021.

---

> > ### Comment · Reviewer_DNNj · 2024-08-08
> > **Response to Author Rebuttal**
> >
> > Thank you for the rebuttal, it satisfied most of my concerns.
> >
> > One remaining question: I am still confused about M and N. Are they learned feature maps (optimized)? Perhaps you can visualize them as you did C, so the reader has a better understanding of what these maps look like (probably good for supplemental material).
> >
> > A few follow-ups for the final camera-ready paper (if accepted):
> >
> > 1) I think it is important that the paper acknowledge that the Hadamard product decomposition is inspired by other papers (such as [1-3]) by explicitly putting those references and sentence into a revised version of the paper.
> >
> > 2) Visualizing C in the main paper greatly assists the reader in understanding what this map is. I would endeavor to include it in one of your figures, or as a small additional figure. Your text description should also be incorporated so that the reader can more clearly understand C. Also if M and N are able to be visualized (see question above), it would be good to point readers to understand those feature maps as well.
> >
> > 3) If the loss function only adds 0.15dB, please revise the language in that section to say a "modest" improvement. It is not a major difference, and do not want to overclaim the contributions of that one loss function. If you want to keep the original language, then I still think a qualitative figure with and without the loss function is needed to convince readers the 0.15dB is significant.
> >
> > 4) For the wall-clock latency time, I think it would be good to breakdown how much time was due to DATUM (which is not your method), and how much was due to your optimizations. That gives the readers some understanding that if they add your module to their base ATM method, this is the expected overhead. Please add this discussion to either the main paper or the supplemental material.

---

> > > ### Author Response · Authors · 2024-08-11
> > > **Further discussion with Reviewer DNNj (denoted as R1)**
> > >
> > > We greatly appreciate your prompt response. Thank you for your follow-up question and suggestions.
> > >
> > > **`Suggestion 2 & Further Concern`** **More details about the feature map M and N**
> > >
> > > Yes, M (Spatial Feature Map), N (Temporal Feature Map), and C (Canonical Spatial Feature Map) are all optimizable during training. M and N are learnable parameters with shapes [H,W,Q1] and [T_res,Q1], respectively. The optimizable parameters in our pipeline include all these feature maps (M,N,C) and two MLPs (content and deformation MLP), as mentioned in line 136 of the main paper. We will emphasize the details of each component by adding more legends and markers to the pipeline figure in the revision.
> > >
> > > Regarding the visualization of M (spatial feature map) and N (temporal feature map), we will include visualizations of the deformation grid (delta x and delta y) and M in the revised version. The delta x and delta y serve as a sampling grid, indicating how each frame is sampled from the canonical field (similar to a keyframe in video compression). This visualization will help clarify the product of M and N for specific x, y, and t coordinates
> > >
> > > **`Suggesion 1`** **Acknowledge the tensor decomposition**
> > >
> > > Thank you for your suggestion. In the revised version, we will reorganize the related work section about INR and add a new subsection to the methods part, explaining the inspiration behind our method's design.
> > >
> > > **`Suggesion 3`** **Description of effectiveness brought by the loss function.**
> > >
> > > Thank you for your considerate suggestion. We will add the qualitative results from our ablation study on loss functions, as shown in the **`attached file figure 4 `**, to the revised paper. This addition will help readers better understand the effectiveness of each component. Additionally, we will revise our language to more clearly explain the effectiveness of each component. We appreciate your feedback, which will improve the clarity and comprehensiveness of our paper
> > >
> > > **`Suggesion 4`** **Running Time Breakdown**
> > >
> > > Thank you for the suggestion. In the revised version, we will include a detailed runtime discussion, breaking down the time for DATUM (0.5 mins) and our method (9.5 mins). Our method is the first refinement strategy for supervised video-based ATM methods struggling with temporal consistency. Therefore, running time efficiency compared to supervised methods is not our primary focus in this work. However, we agree that discussing the runtime of each method, particularly with the breakdown for DATUM, is crucial. This information will help readers understand the computational cost of incorporating our method into existing approaches.
> > >
> > >
> > > We appreciate your timely response to our rebuttal. Please let us know if you have further concerns. We will respond to your comments as quickly as we can

---

> ### Comment · Reviewer_DNNj · 2024-08-11
>
> I am satisfied with these revisions and will raise my rating to weak accept for this paper. The main justification behind my rating is the experimental results shown in the paper (KLT tracking, enhanced temporal consistency) including some moving scenes in the supplemental html page.

---

> > ### Author Response · Authors · 2024-08-14
> >
> > We greatly appreciate your positive evaluation and recognition. All your suggesions will be included in the revised version. Thank you again for your valuable time and review.

---

### Author Rebuttal · Authors · 2024-08-07

**`Shared Question A `** : **How does the proposed method handle large object motion?**

Since our method relies on moderating the temporal regularity of motion in the video, it is natural to ask whether it can distinguish between large object motion and turbulence motion. We provided additional experimental results on large motion cases from the Augmented URG-T Dataset [1], featuring videos with both strong turbulence and fast-moving objects.

As shown in **`attached file Figure 1`**, our approach remains effective for scenes with both large object motion and strong turbulence. This is evidenced by the long KLT trajectories in our restored videos, which successfully capture the original large object motion. Our approach achieves this because its MLP implicitly penalizes temporally irregular motion caused by turbulence while effectively fitting large but regular motion due to the scene itself. This behavior is illustrated in the X-t slices presented in Figure 1; turbulence motion appears irregular (jittering regions), while large object motion, despite being significant, maintains a smooth shape.

Fundamentally, the difference between large object motion and turbulence is that large object motion results in large optical flow magnitude, but the optical flow direction in a local region remains regular. In contrast, turbulence motion causes irregular optical flow directions. Our method (implicitly) allows large optical flows as long as they are locally regular in direction, while surpassing irregular optical flows in a local region. This is why our method can handle large object motion while mitigating turbulence.

[1] Saha, Ripon Kumar, et al. "Turb-Seg-Res: A Segment-then-Restore Pipeline for Dynamic Videos with Atmospheric Turbulence." Proceedings of the IEEE/CVF Conference on Computer Vision and Pattern Recognition. 2024


**`Shared Question B `** : **What is the relationship between turbulence, camera shake, and object motion?**

As discussed in **`Shared Question A`**, in the X-t slice (a way to visualize optical flow over time), turbulence motion appears as an irregular jittering shape due to irregular optical flow direction over time, while object motion, even if large, results in a smooth shape in the X-t slice. Similarly, as shown in the X-t slice **`Figure 2 in attached file`**, camera shake also manifests as an irregular jittering in a local region. Accordingly, by mitigating irregular motion between frames our method can simultaneously compensate for both turbulence and camera shake.

We simulated Brownian motion to represent camera shake/translation and used MiDas [2]  depth to simulate the effect of decreasing translational optical flow inversely proportional to scene depth due to camera motion. In **`Figure 2`**, we present results for scenes with camera shake alone and with a combination of heavy turbulence and camera shake. Our methods effectively mitigate camera shake since it exhibits irregular motion patterns over time. Additionally, for combined camera shake and turbulence, our methods enhance temporal consistency, even when residual camera shake and temporal inconsistencies remain after performing DATUM.

[2] Ranftl, René, et al. "Towards robust monocular depth estimation: Mixing datasets for zero-shot cross-dataset transfer." IEEE transactions on pattern analysis and machine intelligence 44.3 (2020): 1623-1637



**`Shared Question C`** : **Comparison with Unsupervised and Test-time Optimization Baselines.**

We have added additional baselines with two popular unsupervised/test-optimization methods: Mao’s [3] (only designed for grayscale frames) and Li’s [4]. As shown in **`attached file Figures 3`**, both methods are designed for static scenes and fail to capture object motion while removing turbulence. This is evident as both methods replace moving human body parts with an averaged static background.

These unsupervised and test-time optimizations are only effective for static scene turbulence removal and fail with videos containing moving objects. To our knowledge, our method is the only unsupervised and test-time optimization approach designed to address temporal consistency in video turbulence mitigation involving moving objects. This is the key difference and novelty of our method compared to other unsupervised and test-time optimization turbulence mitigation methods.


[3] Mao, Zhiyuan, Nicholas Chimitt, and Stanley H. Chan. "Image reconstruction of static and dynamic scenes through anisoplanatic turbulence." IEEE Transactions on Computational Imaging 6 (2020): 1415-1428.

[4] Li, Nianyi, et al. "Unsupervised non-rigid image distortion removal via grid deformation." Proceedings of the IEEE/CVF International Conference on Computer Vision. 2021.



**`Shared Question D`** :  **Feature Visualization**.

We visualize the canonical image by inputting the canonical spatial feature map into the content MLP without applying $\Delta x$ and $\Delta y$, as shown in the **`attached file Figure 4`**. The canonical image contains most of the video's content, providing a base representation from which other frames can be derived. Consequently, the canonical spatial field functions similarly to a key frame in video compression, serving as a reference for other frames in the sequence to query information.

While directly visualizing the N Temporal Feature Map (dimension [T_res, Q1]) is challenging, we demonstrate its effects through ablation studies. We control the strength of regularization by adjusting T_res. As shown in Table R2-Q3, decreasing T_res achieves better regularization on temporal inconsistency

---

### Decision · Program_Chairs · 2024-09-25

**Decision:**

Accept (poster)

**Comment:**

This paper presents a novel method for improving the temporal consistency of videos affected by atmospheric turbulence, utilizing implicit neural representations (INRs) to model temporal and spatial deformations separately. The proposed framework introduces two key INRs: the temporal deformation field and the spatial content field, both of which are trained on the outputs of a pre-trained atmospheric turbulence mitigation model. The method effectively employs disparity and similarity losses to ensure temporal consistency and maintain content fidelity. Experimental results on both real-world and custom datasets demonstrate significant improvements in temporal consistency, particularly for videos with moving objects, when integrated with state-of-the-art turbulence mitigation models.

The paper is generally well-written, and the majority of reviewers support its acceptance. While reviewer VLLH raised concerns regarding the method's capability to handle large motion and moving objects, the AC believes these concerns are mitigated by the challenging nature of the problem and the promising results already demonstrated in the paper. The ability of the proposed method to effectively address the issue of moving objects has been substantiated by the experiments. Therefore, the AC recommends accepting this paper.